# The Roles of Neutrophil-Derived Myeloperoxidase (MPO) in Diseases: The New Progress

**DOI:** 10.3390/antiox13010132

**Published:** 2024-01-22

**Authors:** Wei Lin, Huili Chen, Xijing Chen, Chaorui Guo

**Affiliations:** 1Clinical Pharmacology Research Center, School of Basic Medicine and Clinical Pharmacy, China Pharmaceutical University, Nanjing 210009, China; 3221092041@stu.cpu.edu.cn; 2Center of System Pharmacology and Pharmacometrics, College of Pharmacy, University of Florida, Gainesville, FL 32611, USA; huilichen@ufl.edu

**Keywords:** myeloperoxidase, COVID-19, cardiovascular disease, cancer, renal disease, neurodegenerative disease, lung disease

## Abstract

Myeloperoxidase (MPO) is a heme-containing peroxidase, mainly expressed in neutrophils and, to a lesser extent, in monocytes. MPO is known to have a broad bactericidal ability via catalyzing the reaction of Cl^−^ with H_2_O_2_ to produce a strong oxidant, hypochlorous acid (HOCl). However, the overproduction of MPO-derived oxidants has drawn attention to its detrimental role, especially in diseases characterized by acute or chronic inflammation. Broadly speaking, MPO and its derived oxidants are involved in the pathological processes of diseases mainly through the oxidation of biomolecules, which promotes inflammation and oxidative stress. Meanwhile, some researchers found that MPO deficiency or using MPO inhibitors could attenuate inflammation and tissue injuries. Taken together, MPO might be a promising target for both prognostic and therapeutic interventions. Therefore, understanding the role of MPO in the progress of various diseases is of great value. This review provides a comprehensive analysis of the diverse roles of MPO in the progression of several diseases, including cardiovascular diseases (CVDs), neurodegenerative diseases, cancers, renal diseases, and lung diseases (including COVID-19). This information serves as a valuable reference for subsequent mechanistic research and drug development.

## 1. Introduction

Myeloperoxidase (MPO), a member of the heme peroxidase enzyme family, is predominantly found in neutrophils, with only a small amount present in monocytes, which is lost during their maturation into macrophages [1,2]. The early recognition of MPO’s presence dates back to 1868 when Klebs observed that guaiac tincture, a substance reacting with peroxidase and used for peroxidase activity assays, could be oxidized by pus, suggesting the existence of MPO in leukocytes. In 1898, Linossier found that H_2_O_2_ is required for the peroxidase reaction in leukocytes [3]. Initially known as verdoperoxidase due to its green color [4], MPO was isolated from leucocytes in 1941 [5]. However, the extraction of peroxidase with a brown-green color from milk indicated its distinction from verdoperoxidase [6]. In 1943, Theorell coined the name MPO, elucidating its source, “myeloid”, and peroxidase activity, “peroxidase” [7].

Due to the bactericidal effects of neutrophils, researchers hypothesized that MPO could also play an antimicrobial role. However, Klebanoff found that MPO, either alone or with semi-lethal amounts of H_2_O_2_, had little bactericidal effect on microorganisms in 1961–1962 [8]. Five years later, Klebanoff proposed and demonstrated that MPO exerted its antimicrobial effect via the MPO-H_2_O_2_-iodide system [9]. In a subsequent study, it was demonstrated that MPO and H_2_O_2_ form antibacterial systems with other halide and pseudo-halide ions. Among these, the MPO-H_2_O_2_-Cl^−^ system exhibited the strongest antibacterial activity [10]. The formation of the MPO-H_2_O_2_-iodide system is as follows: Upon the invasion of pathogens, neutrophils are activated and secrete MPO into extracellular and phagocytic vesicles [11]. Then, MPO takes two electrons from halogen ions and oxidizes them to produce their corresponding hypohalous acids, using the H_2_O_2_ produced by respiratory burst as a co-substrate. MPO, with its powerful oxidation products, is an important factor in the role of neutrophils in innate immunity. Inhibiting MPO results in neutrophils maintaining normal phagocytic activity but with reduced antimicrobial activity [12].

If the antibacterial activity of MPO is not properly terminated, inflammation ceases to be salutary and becomes pathogenic. During oxidative stress or chronic inflammatory processes, MPO is secreted outside the cell, and the concentration of MPO-derived HOCl in the extracellular fluid increases, which leads to the oxidation of DNA [13], RNA, proteins, and lipids [14] due to their strong reactivity with biomolecules, resulting in tissue damage and impaired biological functions [15,16]. Furthermore, these injuries cause inflammation [8], which is thought to be a common mechanism in the pathological processes of many diseases, including but not limited to cardiovascular, respiratory, renal, and neurodegenerative diseases, in which MPO levels in patients’ plasma or other body fluids were significantly elevated and correlated with disease severity [17,18,19,20]. In this review, we focus on the basic information of MPO and integrate extensive information about the relationship between MPO and its active reaction products and multiple diseases, hoping to provide a reference for the development of MPO-related biological targets. In addition, MPO was found to be relevant to the COVID-19 pandemic in the past three years [21,22,23,24]; therefore, the most recent information on the role of MPO in COVID-19 has also been included and discussed in detail in this review.

## 2. Generation and Structure of MPO

MPO was mainly found in neutrophils and monocytes, with a dry weight of 5% and 1%, respectively [25]. The circulating neutrophil count in mice is 10–15%, much lower than that in humans, 60–70%, and the MPO level in murine neutrophils is approximately 10–20% of that in human neutrophils [26]. MPO is actively synthesized in promyelocytes and promonocytes during myelopoiesis in the bone marrow, while it remains inactive in fully differentiated myeloid cells [27]. The specific biosynthetic process of MPO is as follows (as shown in Figure 1) [28]: (a) The production of primary translation, pre-MPO, occurs in the endoplasmic reticulum (ER). Then, after the cotranslational cleavage of the signal peptide and incorporation of high-mannose oligosaccharide side chains, pre-MPO turns into apo-pro-MPO without enzymatic activity. (b) Apo-pro-MPO transiently associates with ER molecular chaperones (including calreticulin and calnexin [29]) in the ER, acquiring a heme group to generate enzymatically active pro-MPO [30]. (c) Subsequently, pro-MPO leaves the endoplasmic reticulum and mostly undergoes intramolecular protein hydrolysis cleavage to form heavy and light chains and form mature MPO dimers that are stored in neutrophils, while a small amount is secreted from the cells [31].

MPO is a dimer with a molecular weight ranging from 120 to 160 kDa [32]. It consists, symmetrically, of a light chain (14.5 kDa, 106 amino acids) and a heavy chain (58.5 kDa, 467 amino acids) [33]. Both of these chains have a biological action and are connected by a single disulfide bridge [34]. MPO is highly glycosylated, which is important for its enzymatic activity. Deglycosylated MPO, on the other hand, exhibits a significant decrease in chlorination activity, low-density lipoprotein (LDL) oxidation, and the ability to produce ROS [35].

The heme group in MPO is the derivative of protoporphyrin IX, with modified methyl groups on pyrrole rings A and C [34]. This modification helps form two ester bonds with the heavy-chain ester bond Glu (408) and the light-chain Asp (260). Heme is attached to MPO via three covalent bonds, in addition to the above two ester bonds, and there is a sulfonate bond between the pyrrole ring A and Met (243) [36], which causes the planar distortion and asymmetry of the heme group, resulting in the unique spectral properties and characteristic green color of MPO.

## 3. MPO-Derived Oxidants

The function of MPO mainly depends on its oxidative catalytic cycle (as shown in Figure 1). The native state of MPO is normally in the ferric form (MPO-Fe^3+^), which reacts with H_2_O_2_ to generate H_2_O and Compound I, which is a ferryl π-cation radical (Fe^IV^ = O) [34]. Compound I can be regenerated into ferric MPO via two different pathways. On the one hand, Compound I can accept two electrons from halide (Cl^−^, Br^−^, and I^−^) or pseudohalide thiocyanate (SCN^−^). In the meantime, it generates corresponding oxidants [37], which are commonly termed the halogenation cycle [38]. The concentrations in the plasma of Cl^−^, Br^−^, I^−^, and SCN^−^ are 100–140 mM, 20–100 μM, less than 1 μM, and 20–120 μM, respectively [10]. On the other hand, Compound I can also be reduced via a two-step one-electron reduction [39] with radicals (nitric oxide, •NO, and O_2_^•−^) or organic compounds (including tyrosine, ascorbate, and steroidal hormones), generating an intermediate, Compound II, which then undergoes a second one-electron reduction to give the ferric species. This process is termed the peroxidase cycle. Native MPO can also directly react with O_2_^•−^ to form Compound III [40], which can be reduced by certain reducing agents [41].

### 3.1. HOCl

H_2_O_2_ is catalyzed by MPO to react with halogen ions, of which 45% reacts with Cl^−^ to form HOCl [10], partly due to the high concentration of Cl^−^ in tissues [42] and its higher reactivity. Also, HOCl is considered the main oxidant product of MPO [43]. The antibacterial activity of HOCl is superior to that of other MPO-catalyzed products [44]. With its short-lived but highly reactive characteristic, HOCl can potentially oxidize most oxidizable groups in most substrates.

However, excessive or misplaced production of HOCl is associated with the onset and development of various pathological processes. HOCl can rapidly react with downstream macromolecules, such as proteins, DNA [13], and lipids [14]. Researchers determined the reactivity of HOCl with potential reactive sites of proteins under physiological pH conditions, showing the following order: Met > Cys ≫ Cystine ≈ His ≈ α-amino > Trp > Lys ≫ Tyr > Arg > Gln ≈ Asn [45].

### 3.2. HOSCN

As the most favorable substrate for MPO (with decreased protonation and a standard redox potential), HOSCN can regulate the final ratio of HOSCN and HOX (X = Cl^−^, Br^−^, and I^−^) produced by MPO [46], and in extreme cases, a high enough amount of SCN^−^ can completely replace other halogen ions. HOSCN can be produced either via the MPO catalysis or direct reaction of SCN^−^ and HOCl [47]. HOSCN can be produced in greater amounts in smokers, who have a higher level of SCN^−^ compared with nonsmokers [48]. The antibacterial ability of HOSCN is inferior to that of HOCl because its oxidizing ability is much weaker than that of HOCl [49]. However, HOSCN can penetrate cells to oxidize intracellular sulfhydryl groups [50].

HOSCN reacts with thiols [51] and selenocysteine [52] in thiol/disulfide-like exchange reactions. As stated above, HOCl can produce irreversible oxidative damage to biomolecules, whereas HOSCN reacts reversibly with less impact on biological functions. The definite effects of HOSCN on diseases are unclear, and some investigators believe that HOSCN can induce cellular dysfunction in some specific cell types, such as macrophages and endothelial cells [53,54], which may be related to the pathological processes of some diseases. Others suggest that the formation of HOSCN is a protective mechanism that competitively inhibits the production of HOCl while consuming H_2_O_2_ [55].

### 3.3. ONOO^−^

When neutrophils are stimulated, respiratory burst occurs with increased oxygen consumption and the production of NADPH oxidase [56,57], and then superoxide (O_2_^•−^) reacts with •NO to form peroxynitrite (ONOO^−^), which contributes to the bactericidal activity of phagosomes [58]. ONOO^−^ further generates a more active oxide, nitrogen dioxide (NO_2_) [59], which mainly exerts its bactericidal effect by oxidizing the sulfhydryl groups of biomolecules [60].

The half-life of ONOO^−^/ONOOH at physiological pH is approximately 1 s, while ONOO^−^ is relatively stable and can react directly with proteins, lipids, DNA [61], and other biomolecules, promoting oxidation and nitration reactions [62], separately or collectively, in some pathophysiological processes [63]. Numerous studies have shown that ONOO^−^ has a close association with some diseases, including but not limited to atherosclerosis [64], myocardial ischemia–reperfusion injury (MIR) [65], stroke [66], sepsis [67] and Alzheimer’s disease (AD) [68]. Currently, there are three primary categories of ONOO^−^ inhibition: first, the use of antioxidants, such as NAC [69] and LA [70], to resist its oxidation; second, the inhibition of the raw materials and enzymes (such as MPO) for ONOO^−^ synthesis; and third, the promotion of the degradation of ONOO^−^, such as via metalloporphyrins and organo-seleno derivatives [71].

## 4. Role of MPO in Innate Immunity

Neutrophils, constituting 60–70% of all human leukocytes [72], employ both oxidative and non-oxidative mechanisms to combat bacteria. Stored in the azurophilic granules and released upon neutrophil activation (triggered by contact with pathogens) [8], MPO plays a major role in the innate immune response [73]. Neutrophil activation by pathogen ingestion triggers the nicotinamide adenine dinucleotide phosphate (NADPH) oxidase enzyme system [74], catalyzing the generation of O_2_^•−^ with O_2_ as the substrate. O_2_^•−^ produces hydrogen peroxide (H_2_O_2_) via a disproportionation reaction, which can react with halides to form the respective sub-halogenated acids catalyzed by MPO [75]. Among these, HOCl, synthesized via the reaction of H_2_O_2_ and Cl^−^, is a very effective component involved in antimicrobial defense and is capable of destroying biomolecules in the cells and tissues of host organisms [39]. HOCl is widely considered to be located in neutrophil phagosomes [76]. An in vitro experiment indicated that HOCI inhibited *E. coli* growth and division rapidly and selectively and impaired its protein synthesis [77]. Researchers found that MPO-knockout mice were more susceptible to bacterial infection, and further experiments indicated that MPO could protect the host against *Klebsiella pneumoniae* [78]. Neutrophils release both MPO and serine proteinases, and their relationships decide the final antimicrobial effect. Low levels of HOCl can inactivate α1-antitrypsin (the specific endogenous inhibitor of neutrophil elastase) [79], while high levels of HOCl can increase the susceptibility of neutrophil-derived proteins to proteolysis [80], implying that the MPO concentration can modulate the immune effect. Pseudomonas infections are more common in patients with MPO deficiency [81]. Population studies utilizing Bayer-Technicon hematological devices have found that a small number of patients with complete MPO deficiency experience severe complications related to infection or inflammation pathologies [82]. However, increased MPO levels are connected to some autoimmune diseases such as eosinophilic granulomatosis [83] and Kawasaki’s disease [84].

The antimicrobial activity of MPO is associated with three main antimicrobial modalities of neutrophils: phagocytosis [85], degranulation, and the formation of neutrophil extracellular traps (NETs). NETs are reticular structures composed of DNA and granular proteins [86], serving as extracellular anti-bacterial structures [87]. However, dysregulated NET formation can also lead to diseases, such as thrombosis [88] and respiratory failure [89]. Pathogens and inflammatory mediators activate the formation and release of NETs [90], which can be divided into three main steps: (a) The activation of NADPH oxidase results in the production of ROS, leading to chromatin denudation. (b) The translocation of neutrophil elastase (NE) and MPO to the nucleus [91] promotes further chromatin unfolding and nuclear membrane rupture [86], which results in the release of chromatin into the cytoplasm, where it is decorated with granules and cytoplasmic proteins. (c) After the disruption of the cytoplasmic membrane, NETs are released with the death of neutrophils. Previous articles have demonstrated that MPO is critical to the formation of NETs [92], as it is a major particle-resident protein that promotes the decondensation of chromatin [93]. In chronic granulomatous disease (CGD) patients who had damaged NADPH oxidase function affecting the production of hydrogen peroxide, it was difficult for the reaction substrate of MPO to form NETs [90]. Also, a study demonstrated that neutrophils from MPO-deficient patients failed to release NETs upon stimulation [94].

Even though MPO plays a pivotal role in the innate immune system, some researchers have differing views on the antimicrobial effect of MPO. It has been observed that MPO primarily targets specific microorganisms, such as *Candida albicans* and *Staphylococcus aureus* [11,27]. Moreover, individuals with MPO deficiency do not seem to experience significant health problems [95]. Some researchers have speculated on this phenomenon, suggesting that other antimicrobial systems may compensate for the long-term absence of MPO at the developmental stage. Additionally, the lack of production of active substances also prevents the inactivation of innate immune molecules within phagocytosed bodies [96].

## 5. Role of MPO in Diseases

### 5.1. Cardiovascular Diseases

Numerous studies suggest a robust association between MPO and CVDs, such as atherosclerosis, MIR [97], coronary artery disease [98], and stroke [99,100,101]. Elevated MPO levels are correlated with a poorer prognosis and increased severity of CVDs [102]. In a long-term clinical trial with 1302 asymptomatic adults, researchers found that adults with MPO at or above the median had greater body mass indices (*p* < 0.001), increased LDL-C levels (*p* = 0.001), decreased HDL-C levels (*p* = 0.001), and elevated systolic and diastolic blood pressure (*p* = 0.001 and *p* = 0.02, respectively) compared with those below the median. Moreover, the CVD event rate was twofold higher in adults with MPO levels at or above the median compared with adults with MPO levels below the median, at 2.3% and 4.6%, respectively [103]. Researchers built an unstable atherosclerotic plaque animal model, showing that the enzyme activity of MPO in unstable plaques was three times higher than that in stable phenotypic plaques. After inhibiting MPO or MPO gene deletion, the unstable phenotype was significantly attenuated, indicating that MPO is a potential therapeutic target for the identification and stabilization of unstable plaque [104]. Then, they collected carotid endarterectomy specimens from 31 patients and coronary plaques removed from 12 patients with cardiac transplantation, and the researchers found that MPO levels were higher in unstable carotid and coronary plaques than in stable plaques [105], providing significant new insights into the role of MPO in vulnerable plaques. MPO disrupts the stability of plaque by reducing the thickness of the fibrous cap, consequently leading to a rise in intraplaque bleeding and an increased likelihood of thrombotic incidents. MPO serum levels were considered as a powerful independent prognostic determinant of clinical outcomes in patients with ACS [18]. Combined with cardiac troponin T, the established prognostic markers of ACS, MPO identified 95% of all adverse events in the c7E3 Anti-Platelet Therapy in Unstable Refractory Angina trial.

Since MPO and its oxidative products are involved in all stages of atherosclerosis, atherosclerosis is highlighted in this review among CVDs (the main mechanisms are shown in Figure 2). A study that detected MPO levels in detergent extracts of atherosclerotic arteries demonstrated that MPO was expressed in human atherosclerotic lesions [106]. Moreover, after using the MPO inhibitor AZM198 in a tandem stenosis model of plaque instability, a study found that inhibiting MPO can stabilize existing vulnerable plaque [107]. Stroke, attributed to vascular disease, was induced by MPO by destroying the integrity of blood vessels, either directly or indirectly [108].

MPO is involved in CVD processes in many ways, contributing to the oxidation of LDL [109], the impairment of high-density lipoprotein (HDL) function, the reduction in nitric oxide (NO) bioavailability leading to endothelial dysfunction [110], and the activation of matrix metalloproteinases (MMPs) [111]. Additionally, MPO-derived-oxidant HOCl promotes endothelial cell apoptosis and shedding, leading to plaque instability [112]. The combination of these functions of MPO makes it an active mediator in the development of CVDs.

#### 5.1.1. LDL

The oxidative conversion of LDL to atherogenic forms is a key event in cardiovascular disease development [113]. MPO is capable of generating a large number of reactive products, including HOCl, chloramines, tyrosine radicals, and nitrogen dioxide. These oxidative products oxidize the proteins, lipids, and antioxidant components in LDL, leading to the formation of oxidized LDL (oxLDL) [114]. In vitro studies showed that the MPO-H_2_O_2_-Cl^−^ system oxidizes L-tyrosine, leading to the production of 3-chlorotyrosine, which serves as a specific marker of oxLDL. Furthermore, HOCl is an intermediate of this reaction. The detection of 3-chlorotyrosine in human atherosclerotic lesions and isolated LDL from atherosclerotic lesions strongly supports the hypothesis that MPO plays a crucial role in the oxidative modification of lipoproteins [115]. OxLDL can inhibit reverse cholesterol transport by promoting the formation of oxidized HDL (oxHDL), thereby impairing the protective effect of HDL on LDL [116].

The oxidative modification of LDL is an early event in the development of atherosclerosis [117]. OxLDL promotes atherogenesis by promoting cholesterol deposition, converting macrophages into foam cells. Retention in the subendothelial space makes LDL a major target of pro-oxidant oxidation produced by arterial wall cells [118]. OxLDL is more readily taken up by macrophages than natural LDL, potentially leading to the generation of foam cells as well as an enlarged lipid core that exacerbates the stress on the fibrous cap matrix, making atherosclerotic plaques more prone to rupture [119]. Furthermore, when LDL is oxidized to oxLDL, it is recognized by CD36, the scavenger receptor of macrophages, contributing to irregular uptake and the formation of foam cells [116].

#### 5.1.2. HDL

HDL possesses several functions that help protect against CVDs. These functions include cholesterol reverse transport [120], anti-oxidation [121], anti-inflammation [122], protecting endothelial cells, and inhibiting the formation of oxLDL [123]. The major protein of HDL [124], ApoA-1, is the main location where HDL exerts its anti-oxidative effect [125], as circulating levels of ApoA-1 or HDL are negatively associated with CVDs [126]. It is worth noting that dysfunctional apoA1 can have a pro-inflammatory effect [127]. HDL, isolated from atherosclerotic lesions, contains a large number of MPO-modified peptides, such as chlorinated, nitrated, and sulfoxidated apoA-I [128], demonstrating the correlation between the modifying effect of MPO on HDL and the development of atherosclerosis.

Studies indicate that both HDL and apoA1 undergo extensive oxidation by MPO in human atherosclerotic lesions, resulting in decreased ABCA1-dependent cholesterol efflux [129] via the specific chlorination of the tyrosine residue site of apoA-1 [130]. Moreover, MPO-modified HDL forms a tighter connection with MPO when HDL is oxidized, creating a vicious cycle [131]. Modified HDL also competes with natural HDL as a ligand for the scavenger receptor BI, potentially interfering with cholesterol mobilization from peripheral tissues to the liver [132]. Besides the direct oxidation of apoA-I, MPO-derived oxidants modify lipids to produce highly reactive dicarbonates, such as malondialdehyde and isoprenoid adenosine, which can form a covalent bond with apoA-I and reduce cholesterol efflux [133,134]. In addition, MPO can induce the production of apoA-I/apoA-II heterodimers in HDL [135]. These heterodimers impaired wound-healing cell migration, and MPO-mediated HDL was shown to have an impaired endothelial healing function [136]. Taken together, MPO is considered to contribute to HDL dysfunction, which participates in the pathologic process of atherosclerosis.

#### 5.1.3. Endothelial Dysfunction

Endothelial dysfunction, which is considered an early marker of atherosclerosis [137], has been linked to the potential role of MPO [138]. Researchers observed that the inhibition of MPO in a mice model of atherosclerosis reduced both the inflammatory response and endothelial dysfunction [139].

NO is produced by endothelial NO synthase (eNOS), which plays a key role in vascular homeostasis [140]. Insufficient NO can increase arterial oxidative stress and endothelial cell damage. MPO can be localized on the surface of endothelial cells and internalized [141], oxidizing NO and limiting its bioavailability [142], causing impaired endothelium-dependent diastole and resulting in endothelial dysfunction. MPO interferes with the eNOS/NO pathway to reduce eNOS activity [143], affecting endothelium-derived relaxation. In an in vitro experiment, MPO induced endothelial dysfunction by decreasing eNOS Ser1177 phosphorylation [144]. Meanwhile, HOCl-modified proteins were found in endothelial cells overlying atherosclerotic lesions in an immunostaining experiment [145]. When exposed to HOCl, the arterial rings of rabbits showed dose- and time-dependent endothelium impairment [146].

Endothelial glycocalyx (EG), playing a key role in maintaining vascular homeostasis [147], is highly susceptible to structural changes due to charge modifications. MPO has a high cationic charge at physiological pH [8] due to its arginine and lysine residues, while EG has a highly anionic nature [148]. Researchers demonstrated that MPO was bound to EG via the heparan sulfate side chain, leading to the collapse of the EG [149], and this reaction was not related to the catalytic activity of MPO.

While completely inhibiting MPO activity with non-selective inhibitors may affect its antimicrobial activity, selectively targeting extracellular MPO shows promise in eliminating MPO-induced endothelial dysfunction without compromising its bactericidal effect. However, currently, there are insufficient data to support whether specific MPO activity inhibition can still impact endothelial function in vivo.

### 5.2. Neurodegenerative Diseases

MPO plays an important role in neurodegenerative diseases, including AD, Parkinson’s disease (PD), cerebral ischemia, and multiple sclerosis (MS). MPO is a critical inflammatory enzyme and therapeutic target, triggering both oxidative stress and neuroinflammation in the pathological process of cerebral ischemia–reperfusion injury. It induces chloride stress and nitrosative stress. MPO catalyzes the reaction of H_2_O_2_ with Cl^−^ to form HOCl, which causes chloride stress [150]. It also catalyzes the formation of NO_2_^−^ from NO, leading to nitrosative stress [151], leading to protein nitrification and oxidation, lipid peroxidation, oxidative DNA damage, and the activation of MMP [152], which are all related to neurodegenerative diseases.

The microglia in the brain are usually in a “resting state” [153], with MPO mainly found in the microglia of diseased brains [154], whereas normal-brain microglia rarely express this enzyme [155]. Activated microglia release inducible nitric oxide synthase (iNOS), producing NO, which is converted into reactive oxidants such as NO_2_Cl and ONOO^−^, leading to neuronal damage [152].

HOCl triggers apoptosis at low doses and induces necrosis, including that of neurons and astrocytes, which are major components of the blood–brain barrier, at high doses [156]. It can also interact with ATP to interfere with energy metabolic processes [157]. HOCl-mediated activation of MMP induces tight junction degradation, leading to blood–brain barrier catabolism [158]. Elevated levels of 3-chlorotyrosine (a biomarker of HOCl) were found in damaged brain regions in [159].

#### 5.2.1. Alzheimer’s Disease

The deposition of β-amyloid (Aβ) is a major pathological feature of AD [160]. MPO is co-localized with the Aβ protein in the senile plaques of cortical sections of AD patients [161]. The gene encoding MPO was involved in the pathway leading to Aβ deposition. AD patients had increased levels of neuronal expression of MPO [162]. In a meta-analysis, the concentration of MPO in peripheral blood was significantly higher in AD patients than that in healthy controls [163]. MPO polymorphisms have been identified as risk factors for AD [164].

MPO has been widely recognized to stimulate macrophages and cause the production of ROS and inflammatory factors. In a study about how MPO and microglia play a role in neurodegenerative Alzheimer’s disease, researchers found that MPO could cause the production of ROS and TNFα, the leading proinflammatory cytokines in this disease, in and around microglia, inducing neuronal apoptosis and necrosis [165]. In AD, microglia can interact with Aβ, thereby stimulating microglia inflammatory responses, which can lead to neuronal loss [166]. Therefore, MPO is a promising biomarker for AD that can help in the detection and risk stratification of AD patients.

#### 5.2.2. Parkinson’s Disease

Neuronal expression of MPO is increased in the substantia nigra [167] in PD. MPO-deficient mice showed resistance to 1-methyl-4-phenyl-1,2,3,6-tetrahydropyridine (MPTP)-induced neurotoxicity, a PD model [168]. As mentioned previously, MPO is found mainly in the microglia of the diseased brain [169], and microglia are mainly located in the substantia nigra, which is the brain region most susceptible to the influence of PD. HOCl chloritizes dopamine and neuromelanin to produce chloro-dopamine [170] and neuromelanin, which can enter dopaminergic cells via dopamine transporters to poison the mitochondria [171], contributing to selective dopaminergic neuron death in the substantia nigra [172].

#### 5.2.3. Multiple Sclerosis

MS is an autoimmune disease that causes inflammatory damage to the central nervous system [173]. Immunohistochemical detection found that MPO is located in macrophages/microglia in nearby MS lesions, showing the role of MPO in the pathogenesis of MS [174]. Moreover, a study that collected the white matter of nine MS patients and seven healthy controls found that MPO levels were the highest in demyelinated white matter, followed by non-demyelinated white matter, and were lowest in control white matter [175]. This means elevated MPO levels were associated with MS, and MPO may contribute to axonal injury within plaques. However, in the EAE model, which is the animal model for MS, MPO^−/−^ mice showed an increase in MS morbidity compared with wild-type (WT) mice, at 90% and 30%, respectively [176]. The study indicated that MPO may have a protective role in MS, which could be due to immunosuppressive effects. However, in consideration of the different MPO levels in humans and mice (which are six-fold higher in humans than in mice), the balance between its immunosuppressive and pathogenic effects might vary. The MPO GG genotype with the high-expression property was related to higher disability, a secondary progressive course of MS (*p* < 0.05), and the degree of brain atrophy (*p* < 0.05) in [177]. However, in a later study, the researchers did not find an association between MPO and susceptibility to the course and severity of MS [178]. This might be due to ethnic differences. As a result, the real association is not clear and should be investigated in different populations.

### 5.3. Cancers

MPO plays a dual role in tumor progression. On the one hand, MPO is involved in promoting tumor initiation, development, and migration; on the other hand, MPO enhances innate immune action during tumor elimination [179]. Increased levels of MPO have been found in biological samples of cancers, such as serum from lung cancer patients [180], plasma from subjects with gynecologic cancer, the neoplastic tissue of colon tumors [181], and so on. Previous research demonstrated that MPO had gene polymorphisms, the most common of which was MPO-463G, a polymorphism that affects MPO gene transcriptional levels [182]. This polymorphism, located 463 bp upstream of the transcription start site, binds to specificity protein 1 (SP1) [183]. The -463G site enhances the binding to SP1 compared with -463A so that the expression of the G allele is several times higher than that of the A allele [184,185]. Previous studies have shown that -463A is associated with a decreased risk of developing lung cancer, liver cancer, bladder cancer, and ovarian cancer [185], while -463G with a higher mRNA expression is believed to be connected to increased risks of many types of cancers [186].

MPO plays multiple roles in the progression of cancer. Caspase-3 plays a key role in the control of apoptosis, mediating cellular autophagy by participating in a cascade of reactions triggered in response to pro-apoptotic signals [187], while the non-enzymatic function of MPO acts by protecting cancer cells from caspase-3-mediated cell apoptosis [188]. The nitrosonium ion, an oxidative product of NO catalyzed by MPO, reduces caspase-3 activity via the nitroxylation of the caspase-3 thiol group, protecting tumor cells from apoptosis [189]. Meanwhile, MPO has been shown to catalyze the bioactivation of polycyclic aromatic hydrocarbons [190] and aromatic amines [191], leading to the production of carcinogenically active metabolites.

Due to increased exposure to oxidative stress, prolonged inflammation can lead to DNA damage [192] that induces malignant cell transformation [193]. Also, the concentration of HOCl is higher in tumor cells, especially in transformed cells, than in normal cells, which can also explain this viewpoint [194]. HOCl is the main oxidation product of MPO in vivo, and researchers have found that apoptosis could be induced by inhibiting the chlorination of MPO and thus reducing HOCl levels [195], implying a new anti-cancer strategy by targeting HOCl. HOCl has been shown to cause slow but effective DNA damage by breaking hydrogen bonds, which leads to DNA double-strand dissociation [196]. Also, HOCl can contribute to DNA structural changes and chemical modifications of the heterocyclic NH groups of guanosine and thymine [13]. Reactions with these groups result in the formation of chloramine, which can cause the double-strand dissociation of DNA [197].

On the plus side, MPO can protect against cancers associated with serious infections, such as cervical cancer. In a clinical study with 100 invasive cervical cancer patients and 122 healthy controls, the GG genotype was protective against cancer compared with the GA genotype, possibly due to the role of MPO oxidation products in killing HPV-transformed cells [198].

### 5.4. Renal Diseases

Elevated levels of MPO and its biological products are found in a variety of kidney diseases [199], including chronic kidney disease (CKD), pyelonephritis, and glomerulonephritis [200]. In a prospective cohort study of 3872 participants with CKD who were grouped by MPO levels, higher MPO levels were associated with a 10% higher risk for CKD progression (*p* = 0.03) compared with participants who held lower baseline MPO concentrations [201].

In a renal ablation model simulating CKD, researchers compared MPO^−/−^ mice with WT mice for the incidence of nephropathy [202]. Compared with WT mice, MPO^−/−^ mice showed significantly diminished glomerular injury, decreased gene expression of renal fibrosis markers, and reduced renal monocytes and macrophage infiltration. Also, they found that the level of plasma MPO tripled after renal ablation, suggesting a role of MPO in fibrotic remodeling [203]. In a clinical trial, the level of MPO was higher in the CKD group than in the control group and increased as CKD progressed [204]. As previously described, MPO, being a cationic protein, interacts with glomerular anionic sites [205], and MPO was also co-localized with the glomerulus after perfusion. With the infusion of MPO and H_2_O_2_ into the kidney, the urine protein content increased significantly, showing severe glomerular injury and swelling of endothelial cells [206]. MPO and non-toxic concentrations of H_2_O_2_ perfused alone did not show the above, suggesting that the injury was dependent on the action of the MPO-H_2_O_2_-halide system. MPO was considered to induce neutrophil recruitment [92], while neutrophils can produce a series of reactive oxygen intermediates when they are activated, mediating tissue injury with subsequent renal failure [207].

### 5.5. Lung Diseases and COVID-19

MPO and its derived oxidant HOCl may induce lung diseases by causing oxidative stress and inflammatory responses. 3-chlorotyrosine is considered a biomarker in respiratory diseases, including asthma, cystic fibrosis (CF), and chronic obstructive pulmonary disease, which are associated with the accumulation of inflammatory cells and oxidative stress [208]. Neutrophil-derived MPO is thought to be a major source of oxidative stress on the pulmonary airway surface in CF [209]. Researchers also administered MPO inhibitors at different stages of lung tumors in model mice. They found that MPO catalytic activity was present in the early stage of the tumors, the inflammation phase [188]. HOCl inactivates protease inhibitors at the inflammation site, leading to the dysregulation of elastase activity in the lungs, which may inadvertently destroy connective tissue fibers [210]. In the airways of children with CF, a great deal of HOCl was produced, which may oxidize reduced proteins like GSH in the airways to destroy the lung epithelium [211]. Also, HOCl was discovered to induce ROS and have potential roles in the pyroptosis of acute lung injury (ALI) death by using the fluorescent probe technique [212].

Researchers found that the use of SAAE, a syzygium aromaticum aqueous extract, effectively reduced MPO activity, LPS-induced lung inflammation, and MMP-2 and MMP-9 activity in mice [213]. Also, the inhibition of MPO reduced morbidity and oxidative stress in mice with cystic-fibrosis-like lung inflammation [214]. Researchers found that intranasally injecting MPO^−/−^ mice with LPS showed reduced pro-inflammatory cytokines and chemokines compared with WT mice [215]. These results showed MPO is involved in the inflammatory process of lung diseases, which has also been observed in the worldwide COVID-19 pandemic in the past three years.

COVID-19 exploded at the end of 2019, caused by severe acute respiratory syndrome coronavirus 2 (SARS-CoV-2) [216], and seriously disrupted healthcare systems, suppressed economic development, and threatened social stability [217], directly and indirectly. Recent studies have shown a correlation between COVID-19 and MPO, the major particle-resident protein of NETs, as mentioned above [218]. Clinical research on COVID-19 found that levels of MPO were significantly upregulated in the serum of COVID-19 patients compared with healthy individuals, and serum MPO levels in the recovered population were reduced to levels comparable to those in healthy people [21]. Also, another clinical trial focusing on COVID-19-positive patients found that plasma MPO levels in patients were significantly higher than those in healthy controls [219]. In an in-silico analysis to assess oxidative stress gene expression levels in COVID-19 patients, these gene expression levels were evaluated using COVID-19 transcriptomic datasets and single-cell datasets from specimens collected from the broncho-alveolar lavage fluid (BALF), whole blood, and lung autopsies of COVID-19 patients with varying disease severities [220]. The researchers found MPO levels were not only increased in the samples of COVID-19 patients in comparison with healthy individuals but also in those of severe COVID-19 patients compared with asymptomatic COVID-19 patients. These results showed a relationship between MPO and COVID-19 severity and the redox homeostasis dysfunction caused by this pandemic disease. The symptoms caused by COVID-19 are related to the cytokine storm that it triggers: the excessive immune response leads to the excessive activation of neutrophils and the subsequent production of large amounts of MPO, resulting in the series of clinical manifestations of COVID-19 [221]. It seems that maintaining a normal inflammatory response is as important as anti-viral treatment. COVID-19-induced dysfunctions, such as vascular inflammation and platelet aggregation, were associated with NO synthase, suppressed by over-generated HOCl [222]. Increased levels of MPO generated more HOCl, affecting NO synthase and contributing to COVID-19-induced dysfunctions, such as vascular inflammation and platelet aggregation.

In response to SARS-CoV-2 invasion, NADPH oxidase is activated, MPO is then released and enters into the nuclei of neutrophils [91], and, eventually, NETs are created and released extracellularly [223]. As stated above, NETs are at risk of triggering and spreading inflammation and thrombosis when not properly regulated [224]. The levels of DNA-MPO complexes in COVID-19 patients have been shown to be higher than in healthy individuals [22]. NETs are considered to bring about increased pulmonary morbidity [22,225], organ damage, and mortality rates, and the formation of microthrombosis, [226] during COVID-19 infection.

However, some researchers view MPO’s role in COVID-19 differently, emphasizing its antibacterial action [227]. Researchers produced recombinant MPO (establishing a human HEK293 cell line stably expressing recombinant MPO) and demonstrated its ability to kill a broad spectrum of pathogens, including bacteria and fungi with or without drug resistance, suggesting it is a promising antibacterial agent for COVID-19.

## 6. MPO Inhibitors in Clinical Trials

MPO has been proven to play many roles in the pathological processes of diseases, which are shown in Figure 3; thus, the inhibition of MPO has received much attention. According to the different mechanisms of MPO inhibitors, they are divided into irreversible inhibitors and reversible inhibitors (as shown in Table 1).

Irreversible inhibitors can be oxidized by MPO to generate a radical that binds to the active site, finally leading to irreversible inhibition [228]. p-aminobenzoic acid hydrazide (ABAH) was the first irreversible MPO inhibitor [229]. This inhibitor has been proven to be effective in various diseases. For example, treatment with ABAH can improve neurogenesis after stroke in the mice transient middle cerebral artery occlusion model [158]. An important series of irreversible inhibitors based on the xanthine structure were developed by AstraZeneca [230]. AZD3241 (verdiperstat), a first-class, oral, selective, and irreversible MPO inhibitor, might protect neurons by alleviating MPO-induced pathological oxidative stress and inflammatory effects [231]. Although no significant protective effects have been seen in phase III trials for the treatment of multiple-system atrophy, it has demonstrated efficacy in amyotrophic lateral sclerosis (NCT04436510). Several researchers have studied AZD3241 at the animal level for other disease applications, such as ALI [232], and some results have been achieved both in vitro (a reduced distribution of β-catenin in the nuclei of pulmonary microvascular endothelial cells after treatment with AZD3241) and in vivo (decreased lung coefficient and pathology scores in a rat ALI model injected with AZD3241). Also, this inhibitor could help improve the response in immune checkpoint therapy for patients with melanoma and immune checkpoint therapy resistance for melanoma [233]. However, the clinical application of AZD3241 still has a long way to go. AZD4831, another MPO inhibitor, has completed clinical trials in healthy volunteers and patients with CVDs, renal impairment, and HF. A double-blind phase 2a clinical study identified biomarker profiles associated with clinical outcomes in heart failure with preserved ejection fraction (HFpEF) and the levels of these biomarkers were downregulated after MPO inhibitor AZD4831 treatment, indicating that the effective inhibition of MPO is a promising strategy for HFpEF patients [234]. Meanwhile, in another study, there were no new safety or tolerability matters in HFpEF patients when extracellular MPO was inhibited by AZD4831 [235]. A sequential phase 2b–3 randomized clinical trial has been registered to evaluate the effects of AZD4831 on symptoms and exercise capacity in HFpEF patients [235]. PF-06282999, based on thiouracil, was developed by Pfizer [236]. There was a phase 1 study evaluating the safety and pharmacodynamic effects of PF-06282999 using LPS to induce inflammation in healthy subjects; however, the study was terminated early due to the safety of LPS. Depending on ligand-based pharmacophore modeling, several compounds via a virtual screening procedure were obtained in [237]. Among them, MPO-IN-28, containing a guanidinium-based structure, had the strongest inhibitory activity against MPO.

Reversible inhibitors bind to the MPO active site via non-covalent interactions with high affinity and low dissociation rates [228]. Researchers found MPO has a binding site for aromatic substrates, as the hydroxamic side chains of salicylhydroxamic (SHA) acid and benzohydroxamic acid (BHA) can bind to the hydrophobic pocket at the entrance of the heme [238]. Approximately two decades later, based on the effect of hydroxamic acid, three substituted aromatic hydroxamates were developed [239]. Among them, a trifluoromethyl-substituted aromatic hydroxamate, 2-(3,5-bistrifluoromethylbenzylamino)-6-oxo-1H-pyrimidine-5-carbohydroxamicv acid (HX1), was the most potent inhibitor with an IC_50_ of 5 nM. N-acetyl lysyltyrosylcysteine amide (KYC), a tripeptide inhibitor, has been widely used in different pharmacological studies and can suppress the production of HOCl and the oxidation of LDL [240].

To further the understanding of the relationship between MPO’s structure and functions, and with the help of new technologies such as virtual screening, a series of innovative MPO inhibitors are being investigated. The past decades have witnessed a surge in public interest in natural antioxidants globally, and some of them have been shown to be effective MPO inhibitors. Due to the high efficiency and low toxicity of natural compounds, the use of natural MPO inhibitors as adjunctive therapy for anti-inflammatory treatments holds great potential. Recently, natural polyphenols and flavonoids, such as quercetin, were considered to reversibly inhibit MPO activity [241].

**Table 1 antioxidants-13-00132-t001:** Representative MPO inhibitors.

Category	Pharmacophore	Inhibitor	Structure	IC_50_	Pharmacological Effects	References
Irreversible inhibitors	Hydrazide	4-ABAH	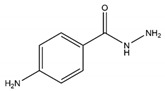	0.3 μM	Improves neurogenesis after ischemic stroke (mice model); improves endothelial function and reduces atherosclerotic plaque development (mice model)	[158,229,242]
	Xanthine	AZD4831	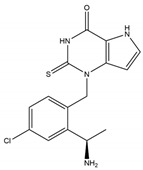	1.5 nM	Downregulates biomarkers associated with HFpEF (clinical trial)	[234,235,243,244]
		AZD3241	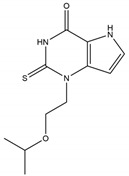	630 nM	Attenuates ALI (mice model); improves PD (clinical trial); and enhances immune checkpoint therapy for melanoma	[231,232,233]
		AZD5904	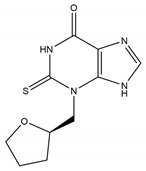	140 nM	Alleviates the relaxation defect in hypertrophic human cardiomyocytes; enhances human sperm function in vitro	[245,246]
	Thiouracil	PF-06282999	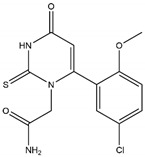	1.9 μM	Promotes atherosclerotic lesion stabilization and prevents atherosclerotic plaque rupture (mice model)	[236,247]
	Guanidine	MPO-IN-28	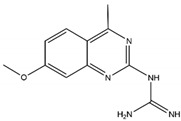	44 nM	Protects against endothelial glycocalyx degradation in primary human aortic endothelial cells cultured with plasma of COVID-19 patients	[237,248]
Reversible inhibitors	Hydroxamic acid	SHA	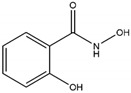	25 μM	No evidence of pharmacological effects; can be used to validate MPO inhibitors in silico	[238]
	Tyrosine	KYC	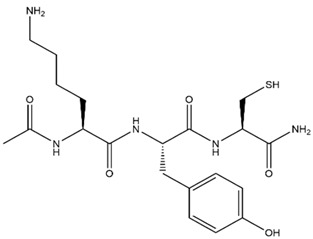	7 μM	Reduces bronchopulmonary dysplasia in hyperoxic neonatal rat pups; reduces oxidative injury and preserves neuronal function in MS (mice model); increases vasodilatation in sickle cell disease mice; promotes brain recovery from injury after stroke (mice model)	[240,249,250,251,252]
	Hydroxamate	HX1	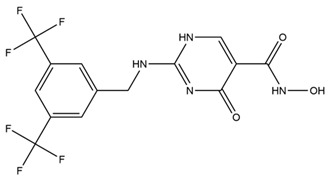	5 nM	No evidence of pharmacological effects	[239,253]

## 7. Conclusions

MPO, as the most abundant protein in neutrophils, plays a crucial role in immune responses via its catalytic cycle. MPO-derived HOCl is a strong oxidant and exerts antibacterial action, which is beneficial for the body in this aspect. However, both MPO and its product can react with biological molecules that cause tissue damage and cell dysfunction; hence, its roles in pathological processes have aroused vast attention. MPO has been implicated in many diseases, including CVDs, renal diseases, lung diseases (including COVID-19), neurodegenerative diseases, and cancers. The precise and detailed mechanisms of MPO’s effects on diseases require further exploration, and understanding these mechanisms will be crucial for the development of MPO-related drugs. Some MPO inhibitors have been found to be effective pharmacologically in in vitro and in vivo studies of these diseases. Clinical trials of some specific MPO inhibitors have been finished or continued based on their clinical outcomes. Despite the absence of MPO inhibitors on the market, the development of MPO inhibitors as investigational new drugs (INDs) is still significant and promising. 

## Figures and Tables

**Figure 1 antioxidants-13-00132-f001:**
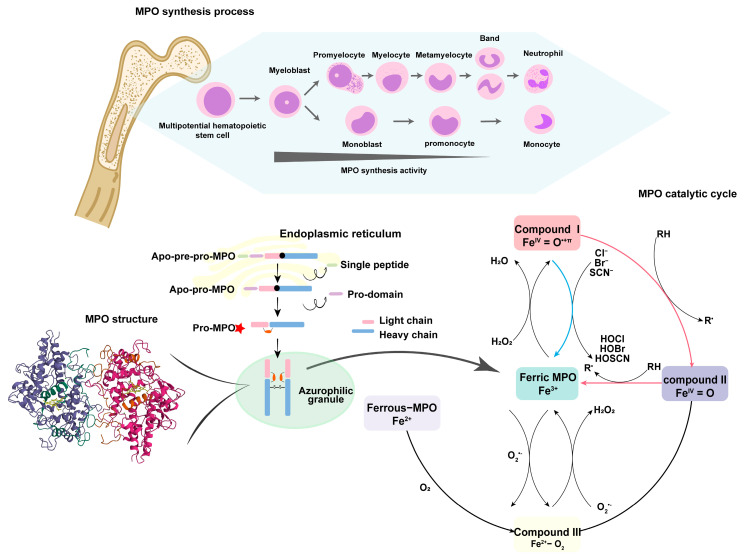
The generation, structure, and catalytic cycle of MPO. The upper part of Figure 1 shows the MPO synthetic process. MPO primarily exists in the azurophil granules of the myeloid series of hematopoietic cells. MPO synthesis occurs at the promyelocyte differentiation stages, and MPO synthesis activity gradually decreases during this process, disappearing in fully differentiated myeloid cells. The first step occurs in the ER, where primary translation production apo-pre-pro-MPO becomes apo-pro-MPO via the cotranslational glycosylation process. The next step, also in the ER, involves apo-pro-MPO combining with ER molecular chaperones and a heme group to turn into enzymatically active pro-MPO. The final step is pro-MPO leaving the ER and entering the azurophilic granule, where the two monomers combine to generate mature MPO. The left side of the picture demonstrates the structure of MPO, constructed in the Protein Data Bank (accession code 7Z53, https://www.rcsb.org/structure/7Z53, accessed on 12 September 2023). MPO (120–160 kDa) is a dimeric enzyme containing a light chain and a heavy chain in each monomer. There is a disulfide bridge connecting the two monomers, and each of them has a heme group that is attached to MPO with two ester bonds (connecting the light chain and the heavy chain) and a sulfonate bond. The right side of the picture illustrates the catalytic cycle of MPO, representing the main ways in which MPO exerts its innate immune function. Ferric MPO reacts with H_2_O_2_ to generate Compound I. Compound I can be backward-reacted into ferric MPO via two pathways, which are marked in blue and red in the figure. Among them, the blue path is called the halogenation cycle: Compound I is reduced by halide ions (or pseudohalide ions) to form ferric MPO and respective hypohalous acids. Also in the red path, Compound I can transform into native MPO via an intermediate, Compound II. Ferric MPO and Compound III can mutually transform by consuming or generating O_2_^•−^.

**Figure 2 antioxidants-13-00132-f002:**
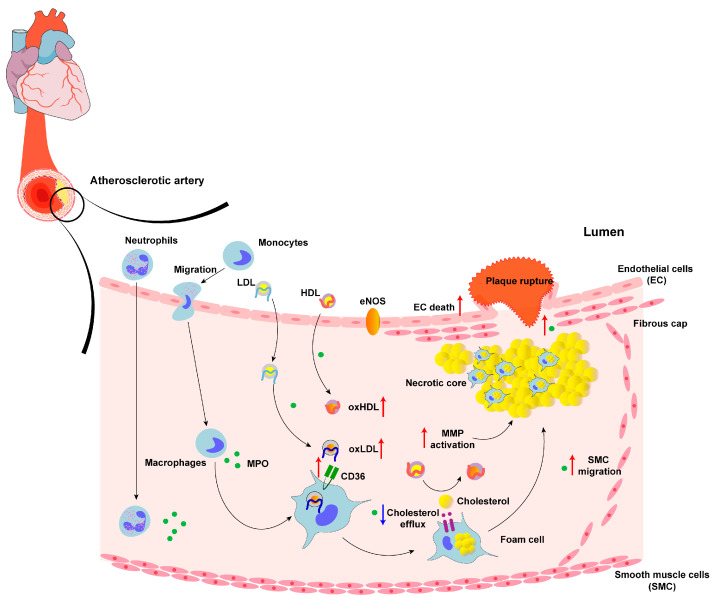
The role of MPO in the pathogenesis of cardiovascular disease. The red arrows demonstrate the process promoted by MPO: the migration of monocytes, the oxidation of LDL, the dysfunction of HDL, the activation of MMP, the migration of SMC, the death of EC, the formation of foam cells, and the rupture of atherosclerotic plaque. The blue arrows show the inhibition process caused by MPO: the reduction in cholesterol efflux and NO bioavailability.

**Figure 3 antioxidants-13-00132-f003:**
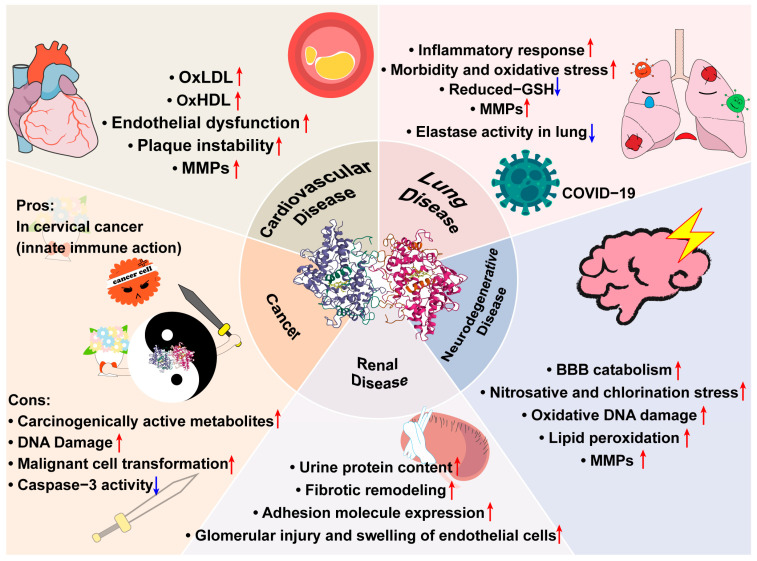
MPO’s various roles in cardiovascular diseases, lung diseases, neurodegenerative diseases, renal diseases, and cancer. (1) In cardiovascular diseases, MPO can oxidize LDL, impair HDL, activate MMPs, as well as increase endothelial dysfunction and plaque instability. (2) In lung diseases, MPO can increase inflammatory responses, morbidity, oxidative stress, and MMPs, while decreasing GSH and elastase activity in the lung. Additionally, MPO is involved in the COVID-19 inflammatory process and is associated with disease severity. (3) In neurodegenerative diseases, MPO can increase BBB catabolism, nitrosative and chlorination stress, and lipid peroxidation as well as activate MMPs and induce oxidative DNA damage. (4) In renal diseases, MPO can increase the urine protein content and adhesion molecule expression and induce glomerular injury and the swelling of endothelial cells. (5) Unlike in other diseases, MPO acts as a two-edged sword in the cancer pathological process. On the one hand, MPO can increase carcinogenically active metabolites, DNA damage, and malignant cell transformation, as well as reduce caspase-3 activity to protect cancer cells from apoptosis. On the other hand, MPO has pharmacologic effects on cervical cancer via its innate immune action.

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
