# Peer review of "The Roles of Neutrophil-Derived Myeloperoxidase (MPO) in Diseases: The New Progress"

_antioxidants, 2024, doi:10.3390/antiox13010132_

Round 1
Reviewer 1 Report
Comments and Suggestions for Authors
This is an interesting review about the role of MPO in several diseases. There are some points which should be re-considered:
1. Abstract: A large part of the abstract describes the action and production of MPO and there is no information how MPO plays role in several diseases, which is the aim of the study.
2. Line 56: "pathological inflammation" what does this mean?
3. Section 4.2 needs re-phrazing because the message is confused
4. Line 229: for "stroke" a reference is required
5. Lines 230-236: Please check if those information are referred to [92]
6. Lines 244-249: The analysis of the findings of ref 18 is not accurate giving a wrong message. Please re-consider it.
7. Lines 251-253, require re-phrasing.
8. Lines 257-258 do not make sense.
9. Lines 301-302: please correct.
10. Lines 312-314: Please rephrase the conclusions of the referred studied
11. Section 5.1 refers to plaque stability and could be better organized reporting first experimental and then clinical data and potential mechanisms. It is a little bit confused now for the reader
12. Lines 469-471 require re-phrasing
Comments on the Quality of English LanguageThe english edition is necessary.
Reviewer 2 Report
Comments and Suggestions for Authors
The authors review the role of myeloid cell derived MPO in several diseases such as cardiovascular neurodegenerative, cancer, renal and lung disease, with nice illustrations, which is of interest.
However, there are major issues to consider:
1. Abstract: The background of six lines should be reduced (too technical for the layperson), but more details on the role of MPO in diseases and potential therapeutic target should be addressed.
2.The section on innate immunity is important and need to be better developed.
3. The different diseases is of interest, but should include more and specific information on the disease processes including ageing and specific, updated citations.
4. The last section on potential drug inhibitors is sketchy, lack a lot of published data needing a through update with a table of the drug candidates.
5. Bibliography has major gaps in the disease section and therapy.
6. The style needs to improved such as the researcher found…. State the published data followed by the reference.
Comments on the Quality of English LanguageImprovement of style.
Round 2
Reviewer 2 Report
Comments and Suggestions for Authors
The revision improved the quality of the review.
Comments on the Quality of English Language
English style and grammar needs to be checked.